# The Role of *luxS* in the Middle Ear *Streptococcus pneumoniae* Isolate 947

**DOI:** 10.3390/pathogens11020216

**Published:** 2022-02-07

**Authors:** Alexandra Tikhomirova, Erin B. Brazel, Kimberley T. McLean, Hannah N. Agnew, James C. Paton, Claudia Trappetti

**Affiliations:** Research Centre for Infectious Diseases, Department of Molecular and Biomedical Science, University of Adelaide, Adelaide, SA 5005, Australia; alexandra.tikhomirova@adelaide.edu.au (A.T.); erin.brazel@adelaide.edu.au (E.B.B.); kimberley.t.mclean@adelaide.edu.au (K.T.M.); hannah.agnew@adelaide.edu.au (H.N.A.); james.paton@adelaide.edu.au (J.C.P.)

**Keywords:** *Streptococcus pneumoniae*, quorum sensing, biofilms, otitis media

## Abstract

The LuxS protein, encoded by *luxS*, is required for the production of autoinducer 2 (AI-2) in *Streptococcus pneumoniae*. The AI-2 molecule serves as a quorum sensing signal, and thus regulates cellular processes such as carbohydrate utilisation and biofilm formation, as well as impacting virulence. The role of *luxS* in *S. pneumoniae* biology and lifestyle has been predominantly assessed in the laboratory strain D39. However, as biofilm formation, which is regulated by *luxS*, is critical for the ability of *S. pneumoniae* to cause otitis media, we investigated the role of *luxS* in a middle ear isolate, strain 947. Our results identified *luxS* to have a role in prevention of *S. pneumoniae* transition from colonisation of the nasopharynx to the ear, and in facilitating adherence to host epithelial cells.

## 1. Introduction

*Streptococcus pneumoniae* colonizes the nasopharynx of up to 97% of infants [1], and frequently this colonization is asymptomatic [2]. However, *S. pneumoniae* can transition from this niche to cause a variety of diseases, including pneumonia upon transit to the lungs, sepsis once in the blood, meningitis upon transit to the brain and otitis media (OM) in the middle ear [3,4]. Persistence of *S. pneumoniae* within these niches relies on its ability to colonize varied host environments. Pertinent to *S. pneumoniae* persistence on mucosal surfaces, such as in the nasopharynx and middle ear, is its ability to form a biofilm community [5,6]. Biofilms exhibit a highly sophisticated structure whereby bacteria can exchange metabolic signals and communicate via a series of small peptide and diffusible autoinducer (AI) molecules. These molecules specifically induce changes in gene expression in target cells in a density-dependent fashion to help the establishment and maintenance of a biofilm. Such interactions are grouped under the general term quorum sensing (QS) systems, or more neutrally as cell–cell signalling systems [7]. 

In *S. pneumoniae*, *luxS*, encoding an s-ribosylhomocysteine lyase (LuxS), is responsible for producing the autoinducer-2 quorum sensing signal [8]. The AI-2 molecule is secreted into the extracellular environment where it is sensed by nearby bacteria and modulates their behaviour. Furthermore, AI-2 plays an important role in biofilm establishment [8]. In the laboratory strain D39, exposure to AI-2 molecule induces biofilm formation, while the deletion of the *luxS* gene results in a reduction in biofilm formation [8]. Interestingly, AI-2 can also increase the virulence of *S. pneumoniae* by shifting metabolic flux towards the catabolism of the sugar galactose, concomitantly increasing capsule production and transit from the nasopharynx to the lungs and the blood [9]. A previous study from our laboratory identified the AI-2 uptake system in Gram-positive bacteria [10]. In particular, we found that the fructose-specific phosphotransferase system (PTS) component FruA is the receptor for the universal QS signaling molecule AI-2 and elucidated the mechanism whereby pneumococci increase the production of capsular polysaccharide and virulence in response to the AI-2 molecule.

Importantly, *luxS* expression has also been shown to be directly related to biofilm formation in clinical isolates [11]. A total of 25 serotype 3 isolates from distinct patients (12 from blood and 13 from ears) were sequence typed (ST) and assessed for biofilm formation and virulence phenotype. All 12 of the blood isolates formed more robust biofilms at pH 7.4. Conversely, all 13 of the ear isolates formed more robust biofilms at pH 6.8. Significantly, pH 7.4 and pH 6.8 correspond to the physiological pH of the blood and middle ear cavity, respectively. Importantly, biofilm formation paralleled *luxS* expression and genetic competence, and the level of *luxS* expression in blood isolates (in conditions optimal for these strains) was higher than *luxS* expression in the ear isolates in optimal conditions for biofilm formation of those strains, indicating that the key role of *luxS* in biofilm formation of isolates from distinct anatomical niches may differ [11]. 

Interestingly, in other bacterial species, *luxS* was shown to have strain-specific functions. In *Haemophilus influenzae*, *luxS* was found to have distinct roles in pathogenesis in isolates obtained from different niches. Strain 86-028NP, which was isolated from the nasopharynx of an otitis media patient, showed reduced biofilm formation and persistence in vivo when *luxS* was deleted [12]. In contrast, isolate R3157, isolated from the middle ear of an otitis media patient, showed no reduction in biofilm formation when *luxS* was deleted; moreover, it showed a hypervirulent phenotype in a chinchilla model of otitis media [13]. 

While a role for *luxS* has been shown in biofilm formation in both *S. pneumoniae* laboratory strain D39 and clinical isolates, an analysis of *luxS* in the pathogenesis of a middle ear isolate has not yet been performed, despite biofilm playing a role in the middle ear pathology. In this study, we aimed to assess a role of *luxS* in the biofilm formation and pathogenesis of the middle ear isolate 947, which we have previously characterized in detail [14]. 

## 2. Results

### 2.1. The Otitis Media Isolate 947 Displays Differences in Expression of Quorum Sensing Associated Genes Compared to the Blood Isolate 4559

We have previously shown that *S. pneumoniae* clinical isolates, which are closely related genomically and belong to the same serotype and multi locus sequence type (MLST), may display distinct virulence phenotypes in mice, in accordance with their original site of isolation in humans (blood versus ear) as well as different capacities to utilize carbon sources [14,15]. Importantly, strains isolated from patients with otitis media are more able to colonize the nasopharynx and utilize galactose as a carbon source, compared to the invasive clinical isolates [14]. Interestingly, we have shown that this capacity is linked to the LuxS/AI-2 quorum sensing system whereby AI-2 signaling via FruA allows the bacterium to catabolise galactose as a carbon source and upregulates the Leloir pathway [10].

To investigate this relationship further, we performed a gene expression analysis on Serotype 14 ST15 blood isolate 4559 and ear isolate 947, for which we have performed a genomic and phenotypic analysis in a previous publication, showing that the distinction in the virulence phenotype is driven by the inability of 947 to catabolise raffinose [14]. Both 4559 and 947 strains were grown to the same OD_600_ (0.2) in chemically defined media with glucose (Glc) as the sole carbon source (CDM + Glc), then washed and resuspended in chemically defined media with raffinose (Raf) as the carbon source (CDM + Raf) and chemically defined media with galactose (Gal) as the carbon source (CDM + Gal), and incubated for a further 30 min [14]. 

In CDM + Raf, expression levels of *luxS* and *fruA* were significantly greater in the blood isolate than in the ear isolate. However, in CDM + Gal *luxS* and *fruA* expression were significantly greater in the ear isolate (Figure 1). These results showed that *luxS* and *fruA* expression were induced by the presence of extracellular galactose in the ear isolate to a greater extent than in the blood isolate. Thus, we further investigated the impact of *luxS* mutation in the ear isolate 947.

### 2.2. Exogenous AI-2 Has Different Effects on the Blood and Ear Isolates

In the human upper respiratory tract, the principal sugar available for *S. pneumoniae* for use as a carbon source is galactose [16]. We have previously shown that supplementation of 10 μM AI-2 partially restored the growth defect of D39 *luxS-* in galactose media [10]. 

We therefore compared the effect of exogenous AI-2 on blood and ear isolates in galactose media. In CDM + Gal, there was a significant increase in the generation time of the blood isolate compared to the ear isolate, but a similar final cell density was observed (Figure 2). Interestingly, supplementation with 10 μM AI-2 increased the final cell density of the blood isolate, but had the opposite effect on the ear isolate, where the final cell density was significantly reduced in presence of AI-2 (Figure 2). Thus, previous observations obtained with the invasive D39 strain [10] are consistent with the role of *luxS* in the invasive blood strain 4559, but opposite effects were found in the ear isolate 947. These results suggest a strain-specific role for *luxS* in *S. pneumoniae*. 

To confirm that the reduced growth of the otitis media isolate 947 in the presence of AI-2 was a dose-dependent phenomenon, we constructed a *luxS-* mutant and assessed the ability of these strains to grow in the presence of different concentrations of AI-2 (Figure 3). In CDM + Gal, there was a marked increase in the generation time and a reduction in the final cell density of the *luxS-* strain compared to the 947 wild type (WT) parent strain. To distinguish between true QS effects mediated by AI-2 from the indirect consequences of *luxS* disruption on the activated methyl cycle, we examined the capacity of various concentrations of exogenous AI-2 to complement the growth defect of *luxS-*. Supplementation with 10 μM AI-2 partially restored growth, but supplementation with lower (4 μM) AI-2 concentrations had no impact whatsoever and higher concentrations (20 or 100 µM) decreased the growth of *luxS-* strain. Interestingly, in 947 WT, the effect of 4 µM AI-2 had no effect on growth, but 10 µM inhibited growth, thereby confirming a dose-dependent effect previously observed in the D39 strain [10].

### 2.3. luxS Is Critical for Adherence of 947 to Epithelial Cells, but Does Not Play a Role in In Vitro Biofilm Formation

To further investigate the role of AI-2 in the otitis media isolate 947, we assessed the ability of the 947 WT, *luxS-* and complemented strains (comp strain) to adhere to the human nasopharyngeal Detroit 562 cell line. The *luxS-* strain had a significantly reduced ability to adhere to the Detroit cells in comparison to both the WT and complemented strain (Figure 4), indicating that *luxS* is potentially important for adherence to the host cell surface. 

As *luxS* has been shown to play a significant role in biofilm formation in the D39 strain [8], we assessed the capacity of the WT, *luxS-* mutant and complemented 947 strains to form a biofilm in vitro. For this assay we used the xCelligence (Agilent Technologies, Santa Clara, CA, USA) biofilm formation system. This new technology is a real-time cellular biosensor, which uses gold microelectrodes fused to the bottom of the surface of a microtiter plate to measure the electric potential across these electrodes. The presence of adherent bacteria at the electrode/solution interface impedes the electron flow. As shown in Figure 5, all three strains showed comparable biofilm formation in glucose and galactose media (Figure 5A,B, respectively), suggesting a different role for *luxS* in this otitis media isolate compared to the laboratory D39 strain. 

### 2.4. luxS Does Not Influence Capsule Production, but Reduces the Capacity of the S. pneumoniae Middle Ear Isolate 947 to Cause Invasive Disease

We have previously shown that *luxS* has a specific role in *S. pneumoniae D39* pathogenesis [12,13]. Therefore, we assessed the infection profile of the 947 wild-type strain, its *luxS* mutant, and its complemented strain, in our intranasal murine challenge model [17]. 

Prior to murine challenge, the capsule was quantified using the FITC dextran exclusion assay, as previously described [18]. No significant differences were observed between the amount of capsule produced by the WT, *luxS-* or complemented strains (Figure 6).

Groups of Swiss mice were challenged with 10^8^ CFU of each strain, and bacterial loads were quantitated in various tissues 24 h post challenge (Figure 7). No significant differences in bacterial numbers in the nasopharynx were observed between the *luxS-* strain and WT-challenged mice. However, the *luxS* mutant was better able to persist in the nasopharynx of infected mice than the complemented strain, with a significantly higher geometric mean (GM) bacterial load (*p* < 0.05) (Figure 7). In the ear compartment, the bacterial loads of the *luxS* mutant were significantly greater than for mice challenged with either the WT or the complemented strains (*p* < 0.01) (Figure 7). A similar trend was also seen in the brain (Figure 7), although this was not statistically significant. Bacterial loads of all three strains in the lungs were comparable.

## 3. Discussion

The LuxS/AI-2 quorum sensing system has been shown to be central to biofilm formation and virulence in multiple bacterial species, including *S. pneumoniae*. In *S. pneumoniae* D39, a *luxS* mutant had an equivalent ability to colonize the murine nasopharynx compared to the WT, but had a reduced capacity to transit to the blood or the lungs [9]. Furthermore, *luxS* has also been demonstrated to be central to the biofilm formation of D39 *in vitro* [8], and adherence to human respiratory epithelial cells [19].

We previously compared and characterized the genomic and transcriptomic profiles of the *S. pneumoniae* blood isolate strain 4559 and the middle ear isolate strain 947 [14], which displayed very distinct *in vitro* and in vivo phenotypes. Interestingly, the ear isolate had a greater capacity to cause OM in mice relative to the blood isolate. OM has traditionally been viewed as a biofilm-mediated disease and thus connected to the LuxS/AI-2 Quorum sensing system. However, direct studies on ear isolates are missing and most of the research on *luxS-*mediated QS has been performed on invasive isolates such as D39. In this context, we endeavored to assess the role of *luxS* in the ear isolate 947. 

Interestingly, we demonstrated the genes involved in AI-2-mediated QS, *luxS* and *fruA*, to have higher expression in the middle ear isolate 947, in comparison to the blood isolate 4559, indicating the potential importance of QS in this strain (Figure 1). The addition of exogenous AI-2 reduced the growth of 947 in galactose media, but enhanced the growth of 4559 (Figure 2). This suggested potential dose-dependent effects of AI-2 in strains with an initially different expression of QS-associated genes, as well as indicating strain-specificity effects of *luxS* in these clinical isolates (Figure 2 and Figure 3). Remarkably, the concentration of the signaling molecule AI-2 required to complement the growth defect of the *luxS-* mutant in the 947 otitis media strain (Figure 3) is the same of that required to enhance the virulence strain D39 [10].

To further understand the role of *luxS* in the middle ear isolate 947, which displayed a higher expression of QS-associated genes, we assessed the ability of the 947 WT, its *luxS* mutant (*luxS-)* and complemented strains to adhere to human epithelial cells, to form a biofilm, and to cause disease in our murine model. 

We have shown that the 947 *luxS-* mutant had a significantly reduced ability to adhere to human respiratory epithelial cells (Figure 4), but did not have attenuated biofilm formation in vitro (Figure 5), suggesting that adherence and potential downstream biofilm formation in this strain may be affected in vivo.

Interestingly, while *luxS* did not influence capsule production (Figure 6), it did impact the infection profile in our murine model (Figure 7). The 947 *luxS-* strain displayed an enhanced transit to the ear, a trend towards increased nasopharyngeal colonization and increased transit to the brain (Figure 7). An increased transit of the *luxS-* strain to the ear and brain in vivo, together with the reduced adherence to epithelial cells, suggests that functional *luxS* may act to facilitate an adherent lifestyle in vivo and to prevent transit to other host niches, characteristic of localized or systemic invasive diseases such as meningitis or otitis media. These results suggested an opposite role for *luxS* in the infection profile of the non-invasive strain 947 in comparison to the invasive isolate D39.

Our results have shown that the role of *luxS* in *S. pneumoniae* is not universal across strains, but rather it is strain-specific, and may be related to the genomic background of the isolate as well as its niche of isolation. Furthermore, 947 had a distinct expression of *luxS-*mediated QS genes compared to the blood isolate 4559, and a distinct impact on infection profile compared to the blood isolate D39. As 947 is a middle ear (OM) isolate, the role of *luxS* in this strain appears to involve the facilitation of adherence and limitation of spread to more invasive sites, whereas in D39, a laboratory strain originally isolated from the blood, *luxS* appears to drive transition to invasive disease. These results imply a need for a re-examination of the role of *luxS* in pathogenesis in a strain-specific manner, which may help to understand the pathogenesis of these strains, and define the role of *luxS* in these processes.

## 4. Materials and Methods

### 4.1. Bacterial Strains and Culture Conditions

The *Streptococcus pneumoniae* serotype 14 ST15 middle ear isolate 947 and blood isolate 4559, were used for initial gene expression experiments. The middle ear isolate 947 was the background used to create the following strains explored in this study: the 947 strains harbouring a mutation in *rpsL* (referred to as WT), the *luxS-* knockout 947 *rpsL* strain (referred to as *luxS-*), and the complemented 947 *rpsL* strain with wild-type *luxS* re-inserted into the *luxS-* strain (referred to as com), were used for this study. The transformation methods used previously for making mutants using the Janus cassette method were used to obtain all mutants used in this study [14]. All bacterial strains were cultured overnight on Columbia agar, supplemented with 5% horse blood, with or without 40 µg/mL gentamicin. Cells were routinely grown in serum broth for murine experiments, or in a chemically defined medium (CDM). CDM is comprised of SILAC RPMI 1640 Flex Media, no glucose, no phenol red (Sigma), supplemented with amino acids, vitamins, choline, and catalase as described previously [20], with either 0.5% glucose (CDM + Glc), 0.5% raffinose (CDM + Raf), or 0.5% galactose (CDM + Gal). 

### 4.2. qRT-PCR Gene Expression Analyses

*S. pneumoniae* 4559 and 947, were grown to the same OD_600_ (0.2) in CDM + Glc, then washed and resuspended in CDM + Raf and CDM + Gal and incubated for a further 30 min. RNA was then extracted, and levels of *fruA* and *luxS* mRNA were then measured relative to gyrase mRNA by quantitative real-time reverse transcription-PCR (qRT-PCR) [15]. Primers used for qRT-PCR are listed in Table 1.

### 4.3. Construction of Mutants

The 947 *luxS-* and complemented strains were constructed using allelic exchange mutagenesis utilizing the Janus cassette, as previously described [17]. This was performed in a 3-step procedure. First, the endogenous *rpsL* in strain 947 (conferring streptomycin sensitivity) was replaced with *rpsL1* allele (which confers streptomycin resistance), by direct transformation. Subsequently, the Janus cassette (harbouring a kanamycin resistant marker and a dominant, counter-selective, *rpsL+* marker) was used to replace the endogenous *luxS* by transformation of a linear PCR product comprising the Janus cassette flanked by 1-kb sequences 5′ and 3′ of *luxS*. Successful transformants were selected for kanamycin resistance and streptomycin sensitivity. Mutants were confirmed by PCR and Sanger sequencing using the primers listed in Table 1 (AGRF, Adelaide, Australia).

To complement the *luxS-* strain, a PCR product harbouring the *luxS* gene amplified from the WT 947, and 1 kb 5′ and 3′ of *luxS*, was transformed into the *luxS-* strain. Successful transformants were selected for streptomycin resistance and kanamycin sensitivity. The presence of the wild-type *luxS* was verified by PCR. 

### 4.4. Assessment of Biofilm Formation

Biofilm formation was measured using the real time cell analyzer xCELLigence (Agilent Technologies, USA), as described previously for *Streptococcus mutans* [21]. This equipment allows for the detection of variations in the impedance signal, due to bacterial attachment and biofilm formation on the gold-microelectrodes present at the bottom of the E-plates (Agilent Technologies, USA). An initial baseline impedance reading of 150 µL of blank media for the sample wells, or 200 µL of media for the controls wells, was taken prior to bacterial cultures being added to wells. Bacteria were initially grown overnight on Columbia blood agar. Bacterial cells were resuspended in CDM+Glc to OD_600_ of 0.2. Fifty µL of bacterial cell suspension was added to the wells of the E-plate for a ¼ dilution. The xCELLigence system harbouring the bacterial cell cultures was incubated at 37 °C for the duration of the experiment. Biofilm formation was monitored over 24 h by recording the impedance signal at 15 min intervals. 

### 4.5. Assessment of Capsule Production

Capsule thickness was assessed using the FITC-dextran exclusion assay previously described [18] and FITC-dextran of 2000 kDa (Sigma, St. Louis, MO, USA). Bacteria were cultured overnight on blood agar, and resuspended in a chemically defined medium (Silac) to an OD 0.7. Then, 80 µL of bacterial cell suspension was added to 20 µL of FITC dextran (10 mg/mL), 10 µL of the resulting suspension was then pipetted onto a microscope slide and a coverslip was applied firmly. Each strain was prepared twice on different days. The slides were viewed using a Olympus FV3000 Laser Scanning microscope with a 60× objective. The mean area of the FITC and Bright Field (BF) of 40–60 colonies of each strain was determined. Images were analysed using Zen 3.5 Software. Statistical analyses were performed using two-tailed Student’s *t* test; *p* values of <0.05 were deemed statistically significant.

### 4.6. Murine Infection Model

Animal experiments were approved by the University of Adelaide Animal Ethics Committee (approval number S-2016-183). Groups of 16 female, outbred, 4–6 week old CD1 Swiss mice were anaesthetized and intranasally challenged with 1 × 10^8^ CFU of the 947 WT, *luxS-* or complemented strains in 50 µL, as described previously. Nasopharyngeal, ear, lung, blood and brain tissue samples were collected 24 h post-infection. Tissue samples were homogenized in 1 mL of PBS, serially diluted and plated onto Columbia blood agar supplemented with 10 µg/mL gentamicin for viable cell enumeration [17]. Gentamicin supplementation specifically allowed for the detection of *S. pneumoniae*, preventing the interference of murine flora in CFU determination. Statistical analyses were performed using two-tailed Student’s *t* test; *p* values of <0.05 were deemed statistically significant.

### 4.7. Adherence Assays

Adherence assays were carried out using the Detroit 562 human nasopharyngeal cell line. Cells were grown in Dulbecco’s modified Eagle’s medium (DMEM), supplemented with 10% fetal calf serum and 1% penicillin and streptomycin, at 37 °C, in 5% CO_2_. The day prior to the inoculation of *S. pneumoniae* to the tissue cultures, wells of the 24-well tissue culture trays were seeded with the Detroit cells (2 × 10^5^ cells per well) in DMEM with 10% FCS. Bacterial cells were resuspended in CDM+Glc at a final OD_600_ of 0.2. 500 µL of bacterial suspension was added to the washed Detroit cells. Concurrently, each bacterial suspension was added in the same volumes to empty wells, to serve as a control. After incubation for 2 h at 37 °C, wells were washed three times with PBS. Following this, cells were detached from the plate by treatment with 100 µL of 0.25% trypsin-0.02% EDTA, and 400 µL of Triton-x100. Samples were serially diluted and spot plated onto Columbia blood agar to enumerate the adherent bacteria. Assays were conducted in triplicate, and with three independent experiments. Data are presented as percentage of adherence relative to the wild-type. Statistical analyses were performed using two-tailed Student’s *t* test; *p* values of <0.05 were deemed statistically significant. 

## Figures and Tables

**Figure 1 pathogens-11-00216-f001:**
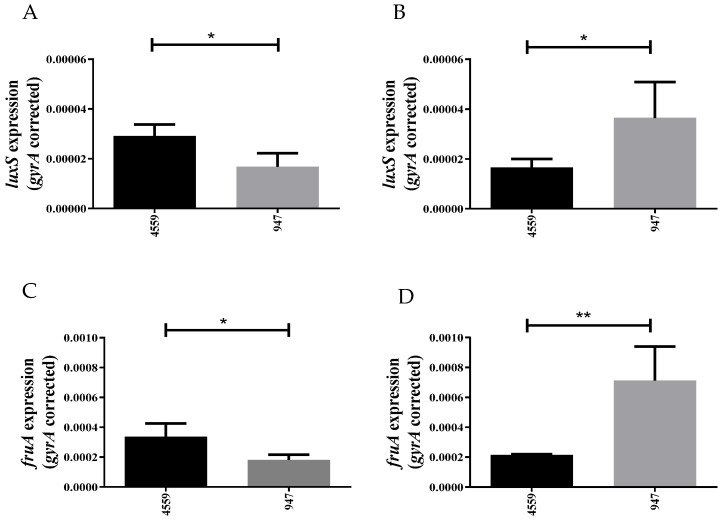
Relative expression of *luxS* (**A**,**B**) and *fruA* (**C**,**D**) in clinical isolates 947 and 4559 grown in CDM + Raf (**A**,**C**) and CDM + Gal (**B**,**D**) was quantitated by qRT-PCR with *gyrA* mRNA as an internal standard. Data are total expression relative to *gyrA* mRNA (mean ± the standard deviation of three independent experiments). *, *p* < 0.05; **, *p* < 0.01 (Student’s unpaired *t* test).

**Figure 2 pathogens-11-00216-f002:**
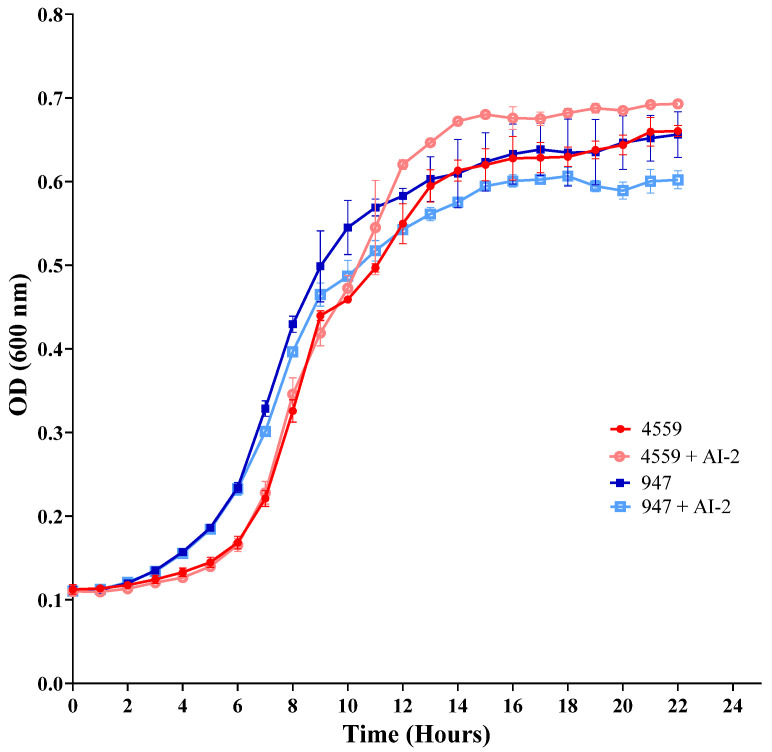
Relative growth of the clinical isolates 947 and 4995, with and without the addition of exogenous 10 µM AI-2 in CDM + Gal over a 24 h period.

**Figure 3 pathogens-11-00216-f003:**
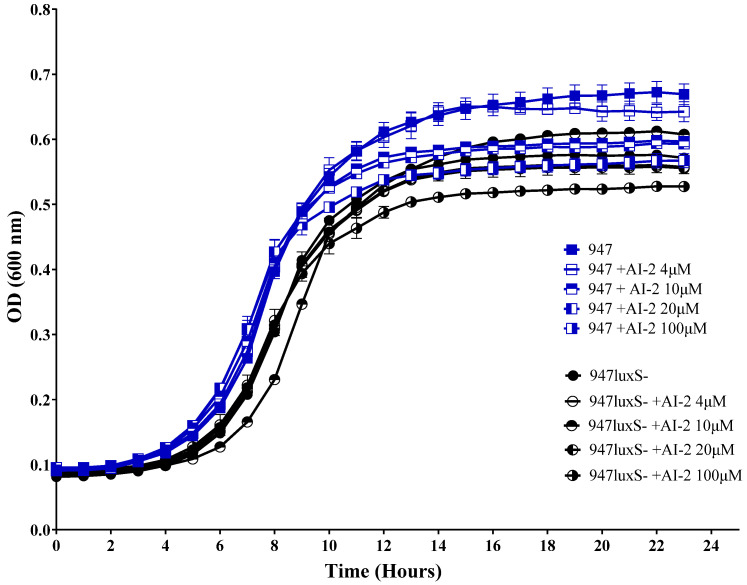
Impact of AI-2 on bacterial growth. 947 and 947 *luxS-* were cultured in CDM + Gal supplemented with 0, 4, 10, 20 or 100 μM and growth was monitored by measuring the OD_600_. Data are mean values of triplicate cultures.

**Figure 4 pathogens-11-00216-f004:**
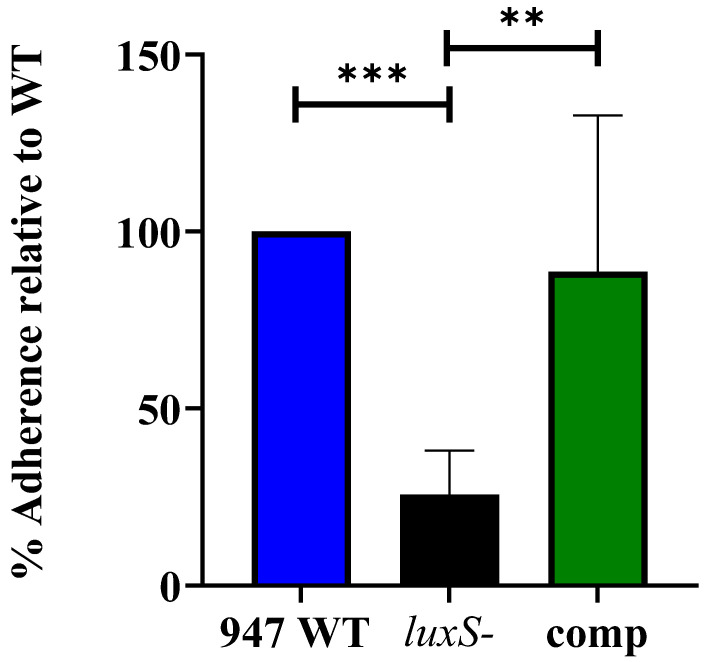
Adherence of the *luxS**-* (black) and complemented (green) strain to Detroit 562 cells, presented as a percentage of the adherence of 947 WT (blue). **, *p* < 0.01; ***, *p* < 0.001.

**Figure 5 pathogens-11-00216-f005:**
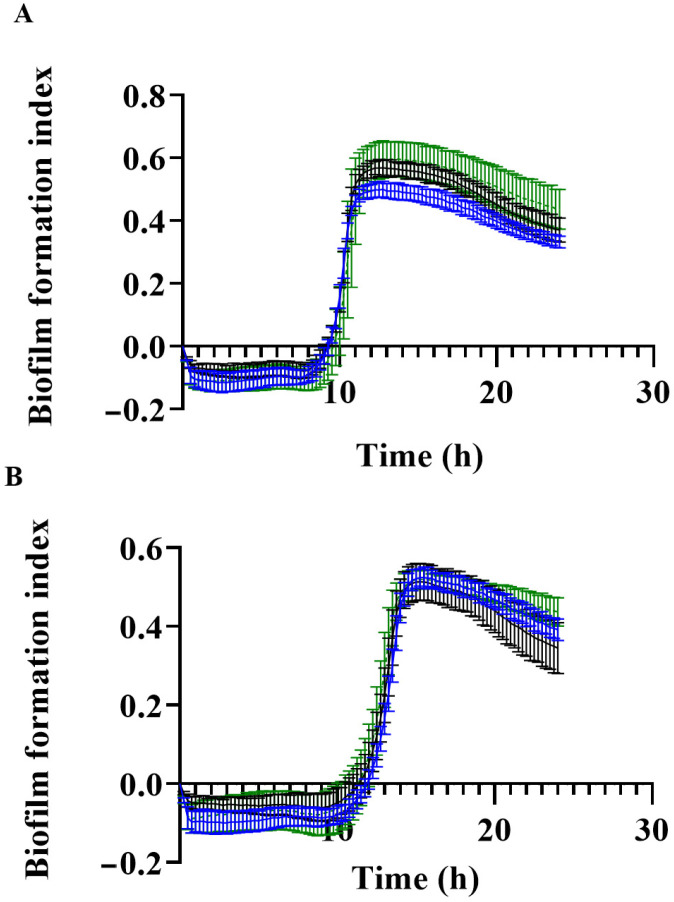
Biofilm formation index of 947 WT (blue), *luxS-* mutant (black), and complemented strain (green) over 24 h in (**A**) CDM + Glc and (**B**) CDM + Gal.

**Figure 6 pathogens-11-00216-f006:**
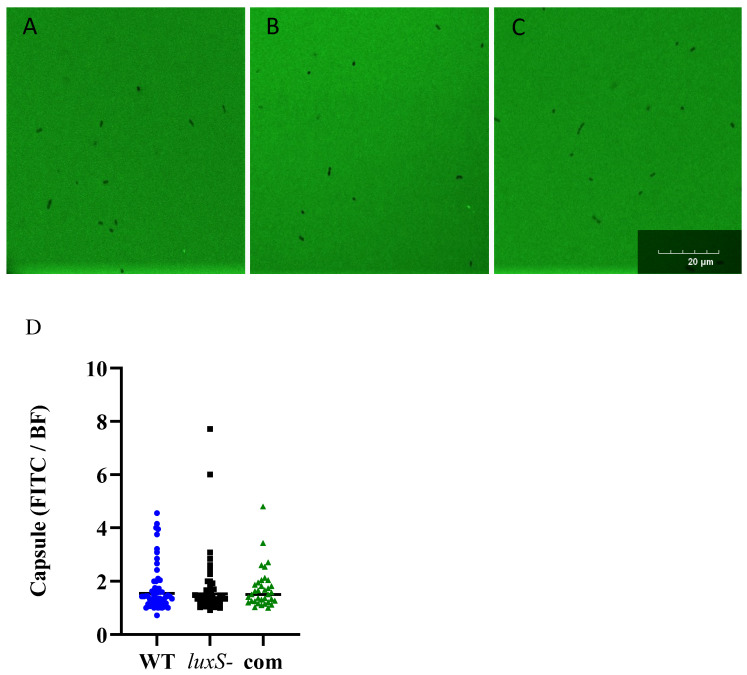
Capsule sizes of strains 947 WT, *luxS-* and complemented strains. (**A**–**C**): FITC-dextran exclusion images of 947 WT(**A**), *luxS-* (**B**) and complemented strains (**C**). All images are to the same scale, taken using a 100X objective, and the scale bar indicates 20 μm. Capsule production by the 947 WT (blue circles), *luxS-* (black squares) and complemented (green triangle) strains, represented as the ratio of FITC to Bright Field (FITC/BF) image areas (**D**).

**Figure 7 pathogens-11-00216-f007:**
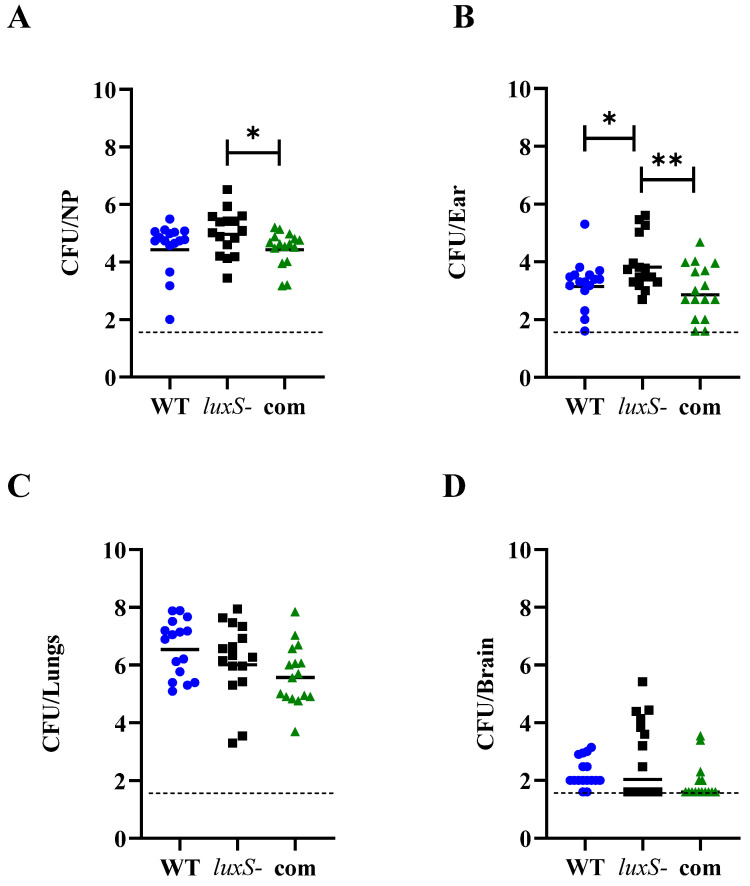
Virulence phenotypes of the 947 WT (blue circles), *luxS-* (black squares) and complemented (com, green triangles) strains at 24 h. Groups of eight mice were infected intranasally with 1 × 10^8^ CFU of the respective strain and at 24 h, mice were euthanized and numbers of pneumococci in the (**A**) Nasopharynx, (**B**) Ear, (**C**) Lungs and (**D**) Brain were quantitated (see Materials and Methods). Viable counts (total CFU per tissue) are shown for each mouse at each site; horizontal bars indicate the geometric mean (GM) for each group, and broken lines indicate the limit of detection. Statistically significant differences in the GM bacterial loads between groups are indicated by asterisks (* *p* < 0.05, ** *p* < 0.01).

**Table 1 pathogens-11-00216-t001:** Primers used in this study.

Primer	Sequence (5′–3′)
gyrA-RT-R	ACCTGATTTCCCCATGCAA
gyrA-RT-F	ACTGGTATCGCGGTTGGGAT
luxS-RT-F	CCCTATGTTCGCTTGATTGGGG
luxS-RT-R	AGTCAATCATGCCGTCAATGCG
fruA-RT-F	TCAATCGTCCAGTTGCTGAC
fruA-RT-R	TTTGTACAAGGCACCACCAA
luxS Janus down	CATTATCCATTAAAAATCAAACGGCTTTCGACAATAACTTCTTTTG
luxS Janus forw	GGAAAGGGGCCAGGTCTCTGCCTTTGAACGTCATGTGATTTAA
luxS forw	CAAATCAATGGCATCAAATCT
luxS down	TGATAAGGTGACAGACTTC
Janus forw	AGACCTGGGCCCCTTTCC
Janus down	CCGTTTGATTTTTAATGGATA

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
