# Peer review of "The Role of luxS in the Middle Ear Streptococcus pneumoniae Isolate 947"

_pathogens, 2022, doi:10.3390/pathogens11020216_

Round 1
Reviewer 1 Report
A few changes were made regarding the order of figures and wording, but generally improved writing in the intro and results would showcase the work more effectively.

Author Response
The review's comments have been integrated in the final paper
Reviewer 2 Report
The study ´The role of luxS in the Middle Ear Streptococcus pneumoniae Isolate 947´by Tikhomirova et al. is an interesting study concerning the role of the luxS gene in the biofilm forming abilities of one selected isolate of S. pneumonia.
The study has been conducted in a well thought manner and the authors present interesting results – it is a pity that the results are limited to one strain. However, the results are certainly of interest to the scientific community and should be published after minor revisions.
Some points needing attention:
English mistakes, typos etc.:
Line 21: ´in´ should be ´is´
Line 149: ´…however, luxS mutant was better able than complemented strain to persist in the nasopharynx of infected mice… ´. Please add 2 ´the´: ´…however, the luxS mutant was better able than the complemented strain to persist in the nasopharynx of infected mice,…´
Line234: there are two commas (,,) after RNA which should not be there
Line 242: ´…was then used the replace the endogenous…´should read ´…was then used to replace the endogenous…´
Line 269: ´… was prepared twice of different days…´ should be ´… was prepared twice on different days…´
Line 286: s. pneumoniae needs to be italic
Line 292: ´… from plate by treatment … ´ should be ´… from the plate by treatment … ´
References:
Line 147 and 151: there is a _ behind the 6 in the reference
Line 228: the reference for ´previously described´ is missing
Line 265: the reference for ´previously described´ is missing
Figures:
In general, the figures would benefit from a color scheme in order to make them easier to understand for the reader.
Figure 4: the figure is difficult to comprehend as it is simply not possible to see the differences between the different treatments; please use 4 different colors in order to improve visibility of the differences
Figure 6: titles of the different figure pats a to d would increase the figure readability; The authors also refer to broken lines in the figure (line 138): it is not clear what the broken lines are.
Missing:
Line 116: company information for the xCelligence system is missing
In paragraph: 4.2. qRT-PCR gene expression analyses a reference to or a table of the primers used is missing.
Line 249: in the description of the PCR the information about the primers used is missing
Line 280-281: please add information about the software used for statistical testing
Line 298-299: please add information about the software used for statistical testing
Line 311-312: the section authors contributions is lacking the information about the author contributions
Other:
Chapter 4.2 is occurring 2 times. Please correct this and the numbering of the following chapters

Reviewer 3 Report
Comments to Pathogens 1537895
The manuscript intends to show the role of luxS in a Streptococcus pneumoniae strain isolated from the middle ear. The result section is rather weak with few data, and there are several flaws in the text and inaccuracies. There is no mechanistic insight. The authors repeatedly compare their data to what was observed previously using 2 other strains. These should have been compared in parallel in the current manuscript. Also, the presence of other TC systems that can compensate for luxS should be presented in the different strains, which might explain the controversy strain-dependent role of luxS. Below are some more specific comments.
Result section
The result section should start with the text and not with the figure. The same throughout the result section. First describe the data and then show the results.
Figure legend 1 should be more descriptive. The y-axis scale is not clear – how can it be in the 0.00001 scale. Usually, real-time PCR is done according to the 2-ΔΔCt method, where the expression of a gene is compared to a control set to 1. Thus, the expression under each growth condition with sugar should be compared to the expression in its absence. The letter size should be larger and readable. 9-47 should be corrected to 947. The Ct of the real-time should be added to Method section.
MLST should be defined first time mentioned.
Line 70: Remove "it".
The Leloir pathway should be described in the introduction.
Line 76: correct to "catabolize".
If 947can't catabolize raffinose, how can you explain the effect of raffinose shown in Figure 1.
Line 78: Define the abbreviations and the rational for the experimental setup.
Line 78: The conclusion that luxS is important for galactose metabolism can't be drawn from the experiment shown in Fig. 1. A deletion mutant needs to be used to show the contribution of luxS.
Figure legend 2 should mention the concentration of the AI-2 and how the kinetic study was performed. Line 89 – there is an extra hyphen. In line 99, there is an extra point. The effect of AI-2 is only seen after longer incubation times. How can this be explained? Can the effect of AI-2 be due to antagonism on endogeneous AI-2? A dose-dependent effect needs to be performed. 10 μM is a quite high concentration. Also, the effect of AI-2 should be done on luxS deficient strains.
A reference should be provided for the sole carbon source galactose in line 92. How was this determined? Why aren't other carbon sources available in the upper respiratory tract?
The conclusion of luxS in lines 99-102 can't be drawn from the experiment using AI-2. How does AI-2 affect luxS expression in each strain? You need to use a ΔluxS mutant to show the contribution of the luxS. Also, the expression of the sensor receptor kinase should be shown.
Figure 3: The effect of luxS deletion on various adhesion molecules should be shown. The effect of exogenous added AI-2 to the luxS mutant should be shown. The data of the 4559 and D39 strains should also be shown in luxS mutants versus wt strains.
Figure 4: Also conventional method of biofilm formation should be presented.
Title in line 127 does not conform with the results presented. In Figure 6, the luxS mutant showed increased CFU. The scale of the y-axis lacks the log description.
Line 138 – Please define GM.
Lines 147 and 151 – there is an extra underline after number 6.
Data for the conclusion in line 157 should be shown.
Since the authors repeatedly compare between the different strains in the text (but show mainly the data of 947), the effect of ΔluxS in each strain on various adhesion molecules should be shown. It could be that other TCSs are more pronounced in the D39 strain, that can explain the controversial role of luxS.
Line 187 – the different expression of QS-associated genes should be provided in the introduction.
Material and Methods:
4.1. The exact composition of the CDM should be provided.
4.2. The exact method of RNA extraction should be provided, and the primer sequences used should be shown. In result section, it was stated that 16S rRNA was used as internal standard, whereas here the gyrase gene was stated. Also, a mistake in line 234: written "gyrase rRNA". Please correct.
4.2. appears twice..
"Com" should be written in full wordings.
The sequences used for mutans construction should be provided. The Janus cassette should be illustrated.
4.3. The biofilm assay seems to be a "non" or "all" detection, and not a gradual detection. A kinetic biofilm study ought to be performed using conventional methods.
4.4. The composition of the Silac CDM should be provided. It seems that (10 mg/ml) is not logical. Please check. Why was not flow cytometry performed? Please provide images of the FITC-dextran binding.
4.5. Please state the concentration of gentamicin? What is the rational of using gentamicin?
4.6. Please state how many Detroit cells were seeded.
Round 2
Reviewer 3 Report
The manuscript is still rather sparse, with only description of a phenotype of a Streptococcus pneumoniae strain isolated from an ear infection whose luxS affects differentially the behavior of the bacteria in comparison with a strain isolated from the blood. The only gene that was analyzed was fruA that they found is affected by luxS in response to galactose versus raffinose. Since they see a really difference of the adherence to Detroit cells, genes involved in the adherence process affected by luxS should also be studied under the experimental conditions used. There is an urgent need for a mechanistic explanation for the differential effect of luxS. Without this, the manuscript quality is not sufficient for publication. In the introduction, the authors speak about different effect of pH on biofilm formation, where the ear isolates can only form biofilm at a pH of 6.8, and not at pH 7.4 (lines 40-43). However, in the manuscript they used RPMI that has glucose and a pH 7.4. How can you explain the discrepancies?
Minor comments.
Line 29 – correct misspelling (illicit) to elicit.
Line 39 – Define ST.
Line 78-79. Please define any abbreviations first time used (CDM, Glc, Raf, Gal). The rationale for these experiments need to be described.
Figure legend 1: the genes should be in italics, also in the figure itself. The asterisks should be made larger.
The importance of fruA in LuxS signaling should be described in the introduction.
Line 94: This line should be removed.
Figure 2: It should include CDM without Galactose, +/- AI-2. Also, the growth curve of CDM with raffinose should be shown with +/- AI-2. The symbols used in the figure need to be improved. Currently it is difficult to see the difference between the open and closed symbols. Maybe because the line transverses the open symbols.
Line 110 – The study was done for 22 hrs according to the graph, such that the text in the legend should be corrected accordingly.
What is the reason for the different behavior of blood versus ear isolates?
Figure 4. The images showing the adherence should also be shown. This can be done by using CFSE-labeled bacteria.
Figure 5 should be complemented with standard biofilm assays that can be done in a kinetic study (e.g., MTT, CV, CLSM). How does luxS affect EPS production?
Line 148: Remove the wording "Figure 6".
Figure 6A-C – It is difficult to see the bacteria. Please improve these images.
Line 168: Correct to (FITC/BF).
Figure 7: The X-axis should be in the same order as the other figures: WT, luxS-, Complementation.
Line 189 – Please write OM in full wordings.
Line 276: Correct to: in a 1/4 dilution.
Line 264: Please describe how the transformation was done.
Line 285: Add a space before 10.
Line 308: Please add the number of cells.
There are some bold letters in the text that should be in plain letters.
Round 3
Reviewer 3 Report
The introduction has been improved. The authors insist not to do some essential gene expression studies that might explain the reduced adherence of the luxS mutant, which is the most profound effect of the mutant. The other effects described for this mutant are rather minor.
There are some missing information (e.g., Figure 3 is not present in the manuscript).
Minor comments:
Line 93: The point after min should be removed.
Line 93: The space between (14) and the point should be removed.
Lines 96-99: Should be removed. The Figure legend appears below Figure 1, lines 107-110. The A-D letters in Figure 1 have moved. Please rearrange them.
Line 113: Remove extra space before (16).
Figure 12 – Please enlarge the letters to readable size. As mentioned before, please make the labels of the bacteria grown in the presence of AI-2 different from those grown in its absence.
Figure 3 is lacking in the text.
Concerning the chemical defined medium it is needed to state the source of RPMI-1640, and to state that the glucose-depleted medium was used.
Also there are extra spaces in several other places in the manuscript.
